# Radon Exposure and Neurodegenerative Disease

**DOI:** 10.3390/ijerph17207439

**Published:** 2020-10-13

**Authors:** Silvia Gómez-Anca, Juan Miguel Barros-Dios

**Affiliations:** 1Department of Preventive Medicine and Public Health, University of Santiago de Compostela, 15782 Santiago, Spain; silvia29896@gmail.com; 2Department of Preventive Medicine, Santiago de Compostela University Teaching Hospital Complex, Santiago de Compostela, 15706 Santiago, Spain; 3Center for Biomedical Research on the Network of Epidemiology and Public Health (Centro DE Investigacion Biomédica en Red DE Epidemiología Y Salud Pública), 15706 Santiago de Compostela, Spain

**Keywords:** radon, multiple sclerosis, amyotrophic lateral sclerosis, Alzheimer, Parkinson, neurodegenerative diseases

## Abstract

*Background:* To carry out a systematic review of scientific literature about the association between radon exposure and neurodegenerative diseases. *Methods:* We performed a bibliographic search in the following databases: Pub med (Medline), Cochrane, BioMed Central and Web of Science. We collected the data by following a predetermined search strategy in which several terms werecombined. After an initial search, 77 articles were obtained.10 of which fulfilled the inclusion criteria. Five of these 10 studies were related to multiple sclerosis (MS), 2 were about motor neuron diseases (MND), in particular amyotrophic lateral sclerosis (ALS) and 3 were related to both Alzheimer’s disease (AD) and Parkinson’s disease (PD). *Results:* The majority of the included articles, suggested a possible association between radon exposure and a subsequent development of neurodegenerative diseases. Some of the studies that obtained statistically significant resultsrevealed a possible association between radon exposure and an increase in MS prevalence. Furthermore, it was also suggested that radon exposure increases MND and AD mortality. Regarding AD and PD, it was observed that certainde cay products of radon-222 (^222^Rn), specifically polonium-210 (^210^Po) and bismuth-210 (^210^Bi), present a characteristic distributionpattern within the brain anatomy. However, the study with the highest scientific evidence included in this review, which investigated a possible association between the concentration of residential radon gas and the MS incidence, revealed no significant results. *Conclusions:* It cannot be concluded, although it is observed, that there is a possible causal association between radon exposure and neurodegenerative diseases. Most of the available studies are ecological so, studies of higher statistical evidence are needed to establish a causal relationship. Further research is needed on this topic.

## 1. Introduction

Neurodegenerative diseases are considered as a heterogeneous group of diseases that affect the central nervous system. They are principally characterized by a progressive neuronal loss. The main risk factor for most neurodegenerative diseases is the age [1]. Although several mechanisms have been proposed to be involved in the development of these diseases, none of them have been identified as the definitive cause [2]. There are more than 100 diseases described as neurodegenerative diseases. However, this review will focus on the most clinically relevant diseases, for which more information is available about the subject of study.

Multiple sclerosis (MS) is an autoimmune disease of the central nervous system (CNS) characterized by a triad based on: inflammation, demyelination and gliosis [3]. The etiology of MS is unknown though, it is believed that it could have a multifactorial origin. The interaction between genetic and environmental factors could trigger an autoimmune attack, which would be responsible for the characteristic damage of myelin and axons in MS [4,5]. Infection with Epstein–Barr virus (EBV) and vitamin D deficiency are the two most studied environmental factors related to MS [6,7]. Furthermore, it is often reported that there is a latitudinal gradient in relation to the prevalence of MS, where the regions close to the equator would have lower values as ompared to the regions with the highest latitude [8].

Amyotrophic lateral sclerosis (ALS) is a neurodegenerative disease characterized by a progressive loss of motor neurons, at both levels:Motor cortex (upper motor neurons) and spinal cord (lower motor neurons). A progressive increase in muscle atrophy and weakness are the predominant symptoms of ALS. A respiratory failure is usually the cause of death in ALS patients due to a paralysis in the diaphragm [9].

Alzheimer’s disease (AD) is a degenerative process where a loss of cognitive functions occurs gradually. It is the most common cause of dementia in elderly patients [10]. Several factors have been related to AD development. Age is the most important risk factor [11]. Cerebrovascular diseases (such as hemorrhagic infarctions, cortical ischemic infarctions, vascular diseases or changes in white matter), high blood pressure, type 2 diabetes mellitus, body weight, plasma lipid levels, the presence of metabolic syndrome, smoking, or a history of head injury, would be others risk factors related to AD. Diet, physical activity, or intellectual activity are considered protective factors.

Parkinson’s disease (PD) is a neurodegenerative disease characterized by a neuronal degeneration in the nigra pars compact substantia and an accumulation of α-synucle in cytoplasmic inclusions (called Lewy bodies) [12,13]. PD is the second most frequent neurodegenerative disease after AD [14,15,16]. Among the environmental factors that could influence PD development are: smoking, coffee, tea consumption, and some certain pesticides exposure. Its possible neuroprotective effect is one of the hypotheses by which tobacco is believed to be a protective factor for PD. However, this fact has only been observed in vitro studies [17,18].

Radon was declared a human carcinogen in 1988 by the International Agency for Resesch on Cancer (IARC) after the demonstration of its carcinogenic effect on the lungs. The American Environmental Protection Agency (USEPA: U.S. Environmental Protection Agency) established at 148 Bq/m3 the radon action level. Later, in 2009, WHO lowered this threshold to 100 Bq/m^3^. Residential radon exposure is the second cause of lung cancer, after tobacco, and the first cause of lung cancer among non-smokers. Between 3% and 14% of lung cancers are related to radon exposure [19].Radon is a noble, colorless, odorless and tasteless gas. It is soluble in lipids and tends to accumulate in carbon-rich body lipid tissues such as the brain tissue [20]. It appears in the Uranium-238 (^238^U) decay chain. There are three isotopes that occur simultaneously: radon-222 (^222^Rn) a direct product of radon-226 (^226^Ra)from the ^238^U decomposition series; radon-220 (^220^Rn) a product of thorium-232 (^232^Th) decomposition; and radon-219 (^219^Rn) a product of Uranium-235 (^235^U) decomposition. ^222^Rn is the most significant isotope, comprising 80% of the total. The half-life of ^222^Rnis 3.8 days, while it is 55.6s for ^220^Rn and 3.6s for ^219^Rn. ^222^Rn has two decay products with a shorter half-life (in milliseconds):polonium-218 (^218^Po) and polonium-214 (^214^Po). Both ^218^Po and ^214^Po are a significant source of radiation in humans. When theseRn^222^ decay products disintegrate, they release α particles. This type of particles impacts the lung epithelial cells, leading to genetic and molecular alterations. The accumulated exposure to low concentrations of ^214^Po and ^214^Powould lead to an increased oncogenic risk [19,21,22]. In this way, radon is chemically inactive, although its decay passes through a series of radioactive decay products. Finally, radon converts tolead-206 (^206^Pb),which is stable [23]. It tends to accumulate indoors (homes or workplaces). Its concentration fundamentally depends on the uranium content around the Earth’s crust where buildings are located. However, factors such as insulation, ventilation, or construction materials also affect radon concentration [24]. Radon is denser than air so, its concentration is usually higher in lower floors than in higher floors. The only demonstrated association related to radon exposure is lung cancer. However, this gas exposure could cause other pathologies such as: oropharyngeal cancer, brain tumors, gastric cancer and hematological malignancies [25,26,27,28,29,30].

Neurodegenerative diseases such as MS, ALS, AD and PD have been proposed as a possible result of radong as exposure. The fact that radon is lipid soluble, is the physiological mechanism by which these diseases are believed to develop. When an individual inhales ambient air with radon, most of the inhaled amount of radon is exhaled immediately. However, the small amount of radon that gets trapped in the lungs goes into the circulatory system. This system serves as a vehicle for the radon to reach to different tissues with high lipid content such as: the brain, the bone marrow or the nervous system. In a certain way, these tissues would be damaged due to radon toxicity and radioactivity [31,32].

The theory by which it is believed that radon would be involved in the pathogenesis of MS, establishes that the nerves myelin sheaths would absorb radon, since it is fat-soluble, releasing α particles which cause an irreversible damage of the myelin sheath. Free radicals’ generation would lead to potential oxidative damage that would also damage the lipid portion of myelin [33,34].

The radon decay products have predisposition for neurons (they are neurotropes), where they cause toxicity (neurotoxicity). They are also insoluble in lipids, an important aspect related to the development of these neurodegenerative diseases. One study showed that PD patients had more amount of radon decay products, specifically ^210^Po (α radiation emitting radionuclide) and ^210^Bi (β radiation emitting radionuclide) in the lipid fraction of the white and cortical gray matter of the frontal and temporal lobes; while, in patients with AD, the proportion of these products was higher in the protein fraction. It is believed that the generated free radicals in the radon decomposition process, which facilitate the formation of highly reactive oxychlorides and have a higher affinity for heavy metals, could increase the absorption of ^210^Po and ^210^Bi. The accumulation of free radicals damages astrocytes (nervous system cells with high radiation sensitivity). Furthermore, after analyzing the distribution of free radicals in different post-mortem brain anatomical regions of an 86-year-old patient who had suffered from AD, most of the radiations emitted by these decay products were detected in the amygdala and hippocampus [35,36,37,38]. The olfactory system could be another CNS entrance route from the ambient air for radon. Particles that are less than 100 nanometers in diameter, when inhaled, are mainly deposited in the olfactory bulb. In this way, these particles are transported through the region of the nasal area and the olfactory nerve towards the CNS. This pathway has already been demonstrated for some environmentally polluting substances. The olfactory epithelium increases its permeability during an inflammation and with the aging, causing therefore a greater translocation of environmental agents that pass into the CNS when inhaled. In fact, the development of neurodegenerative diseases, such as AD and PD, has been associated with olfactory bulb dysfunction [39,40,41].

Radon exposure is currently considered a risk factor for lung cancer. However, the association between radon exposure and several diseases is quite controversial. Accordingly, in this systematic review, we aimed toinvestigate if there is a possible relationship between radon exposure and neurodegenerative diseases.

## 2. Materials and Methods

### 2.1. Bibliographic Search of the Studies Included in the Review

A bibliographic search was performed in the following databases: PubMed (Medline), Cochrane, BioMed Central and Web of Science. We used a predetermined search strategy, based on the use of a series of combinations of terms that are listed below.Spanish and English were the languages used in the search. The search period covered the duration from the start of registrations in each database until 20 February 2020.The systematic review structure followed the principles proposed in the PRISMA statement. The search strategy is presented in Figure 1 [42].

### 2.2. Inclusion and Exclusion Criteria

We used the following inclusion and exclusion criteria to select studies forthe systematic review: (a) Main theme: the possible relationship between radon exposure and neurodegenerative diseases. (b) Study design: experimental studies, cohort studies, case-control studies, ecological and/or cross-sectional studies (c) Participants: only studies carried out on the general population are included. (d) Sample size:case studies were excluded (*n* = 1). (e) Quality indicator (QI): studies with a value less than 1.2 were excluded.

### 2.3. Quality Assessment

We elaborated our ownQIin order to evaluate eachof thestudiesincluded in this review.The QIwas composed of three items: study design(SD), sample size(N), and bias control(BC);each of which received a percentage based on its degree of relevance to the quality of the study. These percentages are represented in Table 1. In addition, within each item, different characteristics were assessed. The items were attributeda higher ora lower score depending on their relevance, as it is proportional to the quality. When a study was scored with different items, we applied the following equation based on a weighted sum of the previously described scores (Table 1):
QI = Σ[0.6 × SD + 0.3 × N + 0.1 × BC](1)

The QI of each studyis shown together along with the description of each of the studies included in the review in Table 2.

## 3. Results

### 3.1. Bibliographic Search Results

A total of 77 results were obtained after performing the search in the different databases with the following combinations of terms: “Radon AND Multiple sclerosis” combination yielded 18 studies, 5 of which fulfilled the inclusion criteria; “Radon AND Amyotrophic lateral scleros is”gave 4publications, two of them were included in the review; 23 results were obtained with the “Radon AND Alzheimer” combination, of which only 2 were included in the review. 15 results were obtained with “Radon AND Parkinson” combination, none of which fulfilled the inclusion criteria. This was due to the majority of the results appear in previous researches so, the studies in this section that fulfill inclusion criteria appear together with those of AD; and finally, with the “Radon AND neurodegenerative” combination, a total of 17 results were obtained, and only one of them were eligible to be included in the review. In this way, based on the total results obtained, it was decided to review the summary of the 77 studies. After abstract review, only 10 studies were selected, as they were the only ones that fulfilled the inclusion criteria. Of these 10 studies, as previously specified when explaining the combinations of terms, 5 were related to MS, 2 to MND (specifically ALS) and 3 to AD and PD. Most of the studies were carried out in Europe, specifically in the United Kingdom and Ireland. However, there is also a study from Norway; another from North America, and the most recent one included in this review, which was performed in Brazil. The most common exclusion criteria were the fact of having an insufficient sample size, or being unrelated to the subject under study.

#### 3.1.1. Radon Exposure and MS

The most recent study included in this section is a cohort study published in 2016, where Groves-Kirk by, C.J. et al. analyzed a possible association between residential radon concentration and MS increased incidence, obtaining non-statistically significant data. They performed a retrospective study about MS incidence in known areas of high radon concentration (England and Wales). The MS incidence was investigated looking for patients with MS first diagnosis over several years, between 2005 and 2012. During this period, 1512 MS cases were diagnosed (1070 women and 442 men). 115 of the new cases registered were assigned to one of the ranges that represent an area of high radon concentration. The relative excess risk (RER) for MS was calculated, standardizing the population of England and Wales for each of the radon concentration ranges. The linear regression of RER against the mean radon concentration in each range showed a positive gradient of 0.22 per 100 Bq.m3 (R = 0.25, *p* = 0.0961). In a certain way, the experimental data showed a great dispersion, not being statistically significant for the 95% confidence interval [31].

There were two case-control studies which have studied the possible relationship between radon exposure and MS development. The most recent, published in 2009, evaluated the possible association between radon exposure and MS prevalence, controlling for other identified risk factors. A total of 148 patients, 97 cases (28 men, 69 women) and 51 controls (20 men, 31 women) were included. After choosing cases and controls, participants were surveyed, asking them about various aspects such as: history of skin cancer, severe burns, sun exposure, outdoor work, sunbathing history, educational information, work and residence, MS family history, and time spent in the different rooms and levels of the house throughout the week. The weighted mean time of radon exposure levels at homes was calculated, using it as a variable, to see if cases spent longer time at homes compared to controls. The results were a total of 97.3 h accumulated per week for the cases compared to 101.4 h for the controls but these results were not statistically significant (*p* = 0.45). However, when this information was stratified by gender, the results showed, in a slightly significant way, that women within the control group spent more hours than those in the case group (116.4 vs. 101.1; *p* = 0.05). On the other hand, radon levels were measured at homes of 46 patients (25 cases: 7 men and 18 women; 21 controls: 12 men and 9 women), provided that they had lived in their current home for at least during the last 5 years, prior to diagnosis. The results showed higher values in cases compared to controls, both on the first floor and in the rooms. Regarding the analyzed variables, the results showed a statistically significant increase inMS risk in individuals with a higher educational level (OR = 2.33; 95% CI = 1.10–4.31), as well as in those who drank milk (OR = 2.13; 95% CI = 1.05–4.32). 400 international units of vitamin D, is the estimated content for milk, taking into account that vitamin D had been considered as a protective factor. Other statistically significant observed data were situations such as the fact of living on a farm (OR = 0.34; 95% CI = 0.16–0.70) or the fact of presenting major diseases (OR = 0.49; 95% CI = 0.24–0.9998), decreased the MS risk. In addition, men have less risk of suffering from MS in general, and a fact that was observed is that this population had significantly more exposure years to outdoor work, compared to the female population (45.76 vs. 6.79, *p* < 0.0001) [43]. The other case-control study included in this section was published in 2008 and analyzed^222^Rn concentration and ^214^Bi retention in the subject rooms of MS affects (cases) vs. healthy subjects (controls). The results showed that ^222^Rn concentrations and ^214^Bi retention were higher in MS subjects, compared to controls (^222^Rn concentration (Bq/m^3^) in MS subjects (mean ± SD): 307 ± 411 vs. controls: 126 ± 122; ^214^BiRetention (Bq) in MS subjects (mean ± SD): 333 ± 312 vs. controls 243 ± 144) [46].

A pilot study performed in Northern Ireland in 2003, tried to find out whether exposure to high radon concentrations would increase individuals’ susceptibility to develop MS. Based on the findings of this pilot study, two further studies were performed: One of them was a survey (carried out in MS affected people) where participants were asked about some house features where these individuals had spent their childhood. Among the questions asked were: the dwelling age, the floors´ number in the building, the heating type, or the water supply source. Although the sample size was relatively small, the provided information was interesting. It revealed that these individuals still lived mostly at the house where they had spent childhood. This building was mostly an old one-floor house, with private water supply (spring well) and fire-based heating [47]. The same year, a Norwegian study performed a bivariate correlation analysis, calculating R (Spearman’s correlation coefficient) between MS incidence rates and the following aspects: radon concentration in indoor environments, annual atmospheric magnesium rain, and annual precipitation. After this analysis, insignificant correlation coefficients were found in the northeast and southeast of the country, while in the rest of southern Norway, the coefficients are mainly significant (*p* < 0.01). These coefficients are positive for MS incidence and ^222^Rn rates, and negative for MS and magnesium precipitation and rainfall. In other words, in most of southern Norway, MS rates increase with increased^222^Rn content, whereas they decrease with increased precipitation and magnesium rainfall [48].

#### 3.1.2. Radon Exposure and ALS

In general, ALS is discussed as the neurodegenerative disease for which we study the effect of radon exposure throughout the review. However, the truth is that the reviewed articles generally speak about motor neuron diseases (MND) but, when analyzing the data of these studies, it is observed that more than 90% of MND patients, suffer from ALS in specific. This fact allows the MND mortality rates to be widely used as substitutes for the ALS incidence rates. The two studies included in the review related to these diseases are ecological studies and were carried out in England and Wales (the oldest, 1996) and in the United States (the most recent, 2016).

The ecological study performed in England and Wales, explored the variations between MND mortality rates in this country, in relation to gamma radiation, indoor radon concentration and increased life expectancy. The initial study research indicated that: both radon concentrations and increased life expectancy were significantly associated with high MND mortality rates. However, high doses of gamma radiation would be statistically associated with a decrease in MND mortality rate [49]. On the other hand, the U.S. ecological study, published in 2016, examined the possible relationship between well water use and MND mortality. This aspect is related to radon exposure because air radon inhalation causes most of the exposure to this gas but, a small part of this exposure occurs through the well water. As a result, they found that the age-adjusted death rates in the United States were significantly associated with residential radon levels. In the study, using the age-standardized MND mortality rates, several variables (in addition to residential radon levels) were analyzed: race; rural/urban environment; smoking history and well water use. Correlation analysis was used to identify multivariate potential relationships between the 5 independent variables mentioned above, and the age-adjusted MND mortality rates. There are a large number of interrelationships between the 5 independent variables, where 6 of the 10 possible pairs show *p*-values <0.10. All variables, with the exception of smoking, were directly related to MND mortality rates (*p* < 0.05). In the study, it was also found that only race (% of white subjects) significantly predicts MND mortality rates (*p* < 0.001). The MND mortality rate is reported to increase by 0.0178 deaths for each 1% increase in the white subject population for each state. Similarly, MND death increases by 0.00783 of 105, for every 1% that well water use increases in the population [50].

#### 3.1.3. Radon Exposure and AD and PD

The case-control study performed by Momcilović B et al. in 1999, analyzed the distribution pattern of radon decay products, specifically ^210^Po (α-radiation emitting radionuclide) and ^210^Bi (β-radiation emitting radionuclide). This pattern was analysed in the lipid and protein fractions of cortical gray matter and subcortical white matter of the frontal and temporal lobes of the brains from death people who had suffered from AD and PD. The previously parameters were also analyzed in smokers, which, together with the AD and PD individuals, correspond to the cases in this study. Controls would correspond to non-smoking deaths individuals. The results showed that the radioactivity in the lipid fraction of the cortical gray and in the subcortical white matter, in both temporal and frontal lobes, was higher in PD patients than in controls. Furthermore, this radioactivity increase was lower in the protein fraction, and only significant in the white matter of the temporal lobe. Compared to PD patients and smokers, AD patients had higher radioactivity in the protein fraction. Smokers also presented higher radioactivity in the protein fraction, both in the cortical gray matter and in the subcortical white matter [37]. On the other hand, the study published in 2017, performed in the USA, after analyzing the AD mortality rates in that country, and after correlating them with the total background radiation, which corresponded to radon exclusively, suggested that ionizing radiation could be an AD risk factor. For this, it was used a simple linear regression method, where it was observed that radon background ionizing radiation was significantly correlated with the AD mortality rates (r = 0.467, 95% CI 0.22–0.657; *p* = 0.001). The same result was obtained for total background ionizing radiation (r = 0.452, 95% CI 0.202–0.646; *p* = 0.001). However, no significant correlation was observed in the case of cosmic and terrestrial ionizing radiation. In the multivariate linear regression analysis, the dependent variable was the AD mortality rate, while the independent variables were radon background radiation, hypertension mortality rate, diabetes death rate and age. Results showed that the AD mortality rate was significantly correlated with radon background radiation (β = 0.508, *p* < 0.001) and with age (β = 0.345, *p* < 0.001) [55]. The most recent study, published in January 2020, is the first study which decided to investigate the ^210^Po concentration, a ^222^Rn decay product, in different sample tissues from corpses in a large city, such as the case of Sao Paulo, in Brazil. The study objective was to analyze the α particles levels emitted by ^210^Po in the olfactory epithelium, olfactory bulb, frontal lobe, and in the lung tissues of 30 corpses. Results revealed that ^210^Po levels (Bq/Kg) were significantly higher in the olfactory bulb when compared to the level obtained in the lung tissues (*p* = 0.037), in the temporal lobes (*p* < 0.001), and also greater in the olfactory epithelium (*p* = 0.071). In addition, taking into account sex, women showed higher levels of ^210^Po in the olfactory bulb, in the lungs and in the temporal lobes, but lower in the olfactory epithelium. However, these differences were only significant in the case of the lungs. The association between ^210^Po levels and age was not significant. Regarding the difference between smokers and non-smokers, the result showed that smokers had higher concentrations of ^210^Poin in the olfactory epithelium (*p* = 0.014), in the frontal lobe and in the lungs, although for these last two tissues, the results were not significant. In the olfactory bulb, the differences between smokers and non-smokers were not significant. Relative to the socioeconomic level, the results showed that the individuals belonging to this subgroup, presented higher concentrations of ^210^Po in the samples [51].

## 4. Discussion

This review focuses on the effect of radon exposure on various neurodegenerative diseases, specifically MS, ALS, AD and PD. The majority of the available studies suggested a possible relationship between radon exposure and these diseases. There are studies that did not obtain statistically significant associations, and considering that almost half of the studies included in the review are ecological studies, it can be concluded, that most of the reviewed studies point towards the existence of an association between these two elements.

Regarding MS studies, the retrospective cohort study showed no statistical association. An insufficient sample size, together with several confounding factors (such as vitamin D levels, sun exposure, tobacco use, population migration, or physical trauma), could be behind the achievement of this result [31]. In a similar way to the previous one, in the case-control study performed by Neurenber J. et al., the small sample size is a limiting factor in the conclusions obtained, although, in this case, the study reported statistically significant results for some variables. Another aspect to take into account is the fact that the presence of important diseases is considered as a protective variable. It is possible that it was because the control group choice was made from a neurological clinic, which implies that participants could have previous medical conditions [43]. In the pilot study carried out in Ireland in 2001, the disease prevalence information, together with radon concentration, were obtained by municipality due to confidentiality reasons. This implies that it is very unspecific to say there is a correlation between these two aspects. In addition, study 2 (within this article) has also several limitations. On the one hand, the sample size was limited (*n* = 67), and it must be taken into account that it was not possible to verify the radon levels of the specific moment in which the survey was carried out. It would also be important to review water sanitation (one of survey items), since one of the possible theories on MS occurrence is autoimmune. Modern sanitary facilities such as bathroom and toilet are associated with lower exposure to microorganisms, and therefore, lower immunity. In this way, people studied in this survey, with less modern sanitary facilities, would have greater immunity, which would imply a possible infectious cause in the disease development mechanism. Regarding household fire, the survey did not ask to specify the fuel type used. However, in the case that the fuel used was peat, it would indicate a greater radon release in these houses [47]. In the study carried out in Norway, the epidemiological information was provided by Westlund [56] where, due to the municipalities diversity site in that country, large or aggregate municipalities were used as units to estimate mortality and disability rates of small municipalities (each unit had a minimum size of 10,000 inhabitants). On the other hand, it must be taken into account that radon emanation depends on the combination of various climatic and geological factors. Several studies show that on the Norwegian soil surface, some of the present ions would be cleaned due to precipitation and melting water, or, what is equally likely, to be partially exchanged for some magnesium, or other marine origin. Regarding rainfall, it must be taken into consideration that they increase from the interior to the coast. The chemical properties of radon are similar to those of some of the alkaline ions, such as cesium, barium, rubidium so, data on the tendency of these ions on the soil surface could be extrapolated to radon. In this way, the large amounts of precipitation, with the consequent magnesium rain on the east coast, could be related to low levels of soluble radon in this area, and therefore low exposure to this gas.

Regarding MND ecological studies, it is important to appreciate several aspects. On the one hand, related to the study published in January 1997, it must be considered that there is no evidence of change in the population mobility. Around 80% of the population remained in the same municipality during the 30 years prior to their death, which reveals great population stability. One of the problems of ecological studies, as the case with this one, is the low death number to calculate the MND age adjusted mortality rates. No distinction was made between family or sporadic MND origin. In the initial investigations about radon concentrations and MND mortality association, it seemed that the presence of a small number of municipalities with high radon concentrations could have influenced the results obtained. However, correlations were subsequently carried out, excluding these municipalities that had a high concentration, and even so, the results showed a positive and statistically significant correlation. The difference found in the results, between the γ and radon radiations, could be explained by the different biological processes caused by these radiations. On the one hand, γ-radiation has a high penetrating capacity, unlike α-radiation. Furthermore, γ radiation has a LET (linear energy transfer) [57]. Another important aspect is the fact that the negative effects of α radiationexposure (observed in a wide range of medical conditions) could be increased by a superoxide dismutase dysfunction in MND familiar cases [58]. Furthermore, the positive association between male mortality and radon gas exposure has also been observed in this article, compared to the positive association between female mortality and life expectancy. It is possible that susceptible males are more vulnerable than susceptible females to environmental factors, such as radon gas exposure.In the study performed in the United States and published in 2016, the initial correlation results between MND mortality rates and residential radon levels were similar to those of the previous study that was carried out in England and Wales, between 1981 and 1989. Both studies showed a positive correlation. However, in the case of the Great Britain study, the correlation analysis was not adjusted by other factors, while the most recent study performed in the United States controlled for confounding variables. In this way, the most recent study on this disease shows no longer significance in more sophisticated models including variables such as race or well water use [49,50].

With respectto AD and PD studies, the one published in 1999, observed as a final result, that radionuclides accumulate 10 times more in the brain of individuals who have suffered from AD or PD, and even more in smokers; versus controls, who are non-smokers. These results could be considered quite reliable due to the outstanding statistical analysis performed, since each sample was analyzed several times, and two methods were used to measure radiation (both α and β particles). The composition of fatty acid differs betweenthe cortical gray matter and the white matter. However, the observed radioactivity was approximately the same in these two brain areas [58]. One of these radiations’ consequences, could be the generation of free radicals which would damage the molecular biological structure due to protein oxidation, the lipid peroxidation or the DNA intercalation. Furthermore, they would alter the transduction signal, the cell membrane function, and the gene expression at cellular level. On the other hand, astrocytes and glial cells seem to be the most sensitive brain cellsto radiation. However, neurons, resist more the radiation because they do not divide. Other aspect to consider is that nicotine has a mimetic effect on cholinergic receptors, and smokers have the highest radionuclides activity in the protein fraction of cortical gray matter and subcortical white matter. This canbe related to the fact that nicotine binds to specific protein receptors and hyperpolarizes the membrane temporarily [37]. The ecological study published in 2017 assessedthe association between AD death rate in the U.S. and ionizing radiation, and in specific the study examined the effect ofradon emittedradiation. The results showed a significant effect despite using hypertension, diabetes and advanced age as covariates. These covariates were established due to their known effects on the disease. Radon intranasal inhalation could cause the radiation from this gas to reach the brain and hippocampus, which would initiate AD. Accumulated damage over time makes from aginga risk factor [59]. The estimated amount ofradiation in thebrainis low. However, the relationship between radon exposure and brain tumors has already been demonstrated [48,49,60]. Despite not being able to relate specific radon exposure to this type of tumors at pediatric age, a French study suggests that Υ exposure radiation could be related to the development of pilocyticastrocytomas [61]. In this 2017 ecological study, although we also investigated the contribution that cosmic or terrestrial radiation could have in AD, the results did not show any correlation in the way they did for radon. This could be due to the fact that this type of radiation contributes in a scarce way within the total radiation so, the greater radon contribution could hide the other radiation types effects on AD. Among the study limitations is that the diagnosis reported indeath certificatesmight not be completely correct, especially for those individuals who die at nursing homes or at home. Another limitation is the fact that vascular dementia can be clinically superimposed on AD, and there may be individuals with both diseases [51]. The study performed in Brazil, despite having found higher ^210^Po concentrations in the olfactory bulb, the frontal lobe, and in the lung tissue; the difference in ^210^Po concentration between sexes was not significant. Relative to socioeconomic status, previous studies have already indicated that low social classes individuals are more exposed, and more vulnerable to the most dangerous polluting particles, including radon [19,61,62]. Relative to the analysed smokers’ sample, the statistically significant results showed that these individuals had higher ^210^Po concentrations in the olfactory epithelium, compared to non-smokers. This aspect could be explained by the fact that different fertilizers are used to grow tobacco plants [63]. The results related to gender are consistent with those obtained in previous studies. In this Brazilian study, higher ^210^Po concentrations were obtained in the different analysed tissues (olfactory bulb, olfactory epithelium, frontal lobe, and lung). A study performed in Galicia, which investigated the radon exposure and brain tumor mortality association, also showed a higher risk among women. Consequently, it is suggestedthat women are more vulnerable to the radiation effect than men [29]. The reduced lungs clearance capacity, which is observed in both smokers and in females, could explain the fact that higher ^210^Po concentrations were obtained in the olfactory bulb, compared to the lungs. Regarding the frontal lobe, previous studies, specifically a case study which investigated the brain locations with the highest accumulation ofthese substances, had already reported this. In fact, a study observed that areas such as the amygdala or the hippocampus were also considered targets, and that the ionizing radiation effect on these anatomical regions of the brain could explain the clinical progression of patients suffering from dementia [36]. The main limitations of this study are the small sample size, together with the absence of quantification of individual radon exposure [54].

Finally, it is important to take into account that several of the articles included in the review are ecological studies. Accordingly, it should be considered that one of the limitations of these studies is that the interpretation of the statistical information of the individuals is inferred from the information derived from the group to which these individuals belong. This fact is known as ecological fallacy.

## 5. Conclusions

After conducting this review, we can accept that:

It cannot be concluded that radon exposure is a cause for the development of neurodegenerative diseases.

We can affirm a possible statistical relationship between these diseases and radon exposure.

Furtheranalytical epidemiological studies (Cases-Controls and Cohorts) are necessary to establish this relationship.

## Figures and Tables

**Figure 1 ijerph-17-07439-f001:**
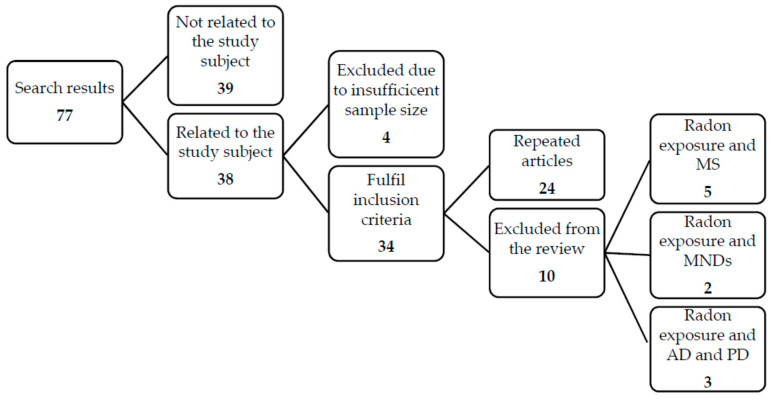
Search summaryforallterms’ combinations.

**Table 1 ijerph-17-07439-t001:** Scores of QI items.

Study Design (SD)	Score
Cross-sectional (CS)	1
Cases and controls (CC)	2
Cohorts (CHO)	3
Experimental (Exp)	4
**Sample Size (N)**	
15 ≤ N < 30	1
30 ≤ N < 50	2
N ≥ 50	3
**Bias Control (BC)**	
Completed	1
No completed	0

**Table 2 ijerph-17-07439-t002:** Description of studies included in the review.

Authores [Ref]	Publication	Publication Year	Country	Study Design	Sample Size	Study Objective	Exposure Characteristic	Statistic Association	QI
Groves-Kirkby, C.J. et al. [31]	J. Environ Radioact	2016	England and Wales	Retrospective Cohort	1512	Investigate the possible association between gas radon concentration at home and the increasing MS incidence	*Radon Atlas for England and Wales*	NO	2.7
Neuberger, J. et al. [43]	MULTIPLE SCLEROSIS	2009	United States	Cases and Controls	148 Cases: 97 Controls: 51	Evaluate the relationship between MS prevalence and radon gas exposure, taking into account other risk identified factors	Arithmetic mean of the Rn (Bq/m^3^) concentration obtained between 1987–1989 from the different rural municipalities aggregates where houses are located [44,45]	YES (Significative association between radon gas exposure and MS prevalence)	2.1
Lykken, G. I. et al. [46]	Journal of neuropathology and experimental neurology	2008	Croatia	Cases and Controls	30 Cases: 15 Controls: 15	Investigate radon exposure as an inductor factor of MS	Canisters	YES (^222^Rn exposure and la ^214^Biretention is higher in the cases)	1.5
Gilmore M. et al. [47]	Environ Geochem Health	2003	Ireland	Study 1:Relationship between MS Ireland members and radon levels Study 2: Survey about MS characteristic houses	*Study 2:* 67	Investigate if some ambient factor, as the radon gas exposure in the infant period, increases the genetic predisposition for MS	Radioactivity map published in 1995 by Appleton and Ball	Not specified	1.5
Bølviken B et al. [48]	Neuroepidemiology	2003	Norway	Ecologic	>100	Investigate radon gas exposure as a risk factor for MS	Radtrack^®^	YES	1.5
Neilson, S. et al. [49]	FC journal of neurology	1997	England and Wales	Ecologic	>100	Investigate the possible relationship between thesourcesof radiation distribution and epidemiological MND mortality data in England and Wales	Rn concentrations were monitored byNRPB (National Radiological Protection Board)	YES	1.5
Gary G. Schwartz et al. [50]	Amyotrophic Lateral Sclerosis and Frontotemporal Degeneration	2016	United States	Ecologic	>100	Investigate the relationship between the MND age adjusted mortality rates in United States, residential radon levels, well water use and other variables using a multivariable analysis.	US EPA (United States—Environmental Protection Agency)	YES (MND age adjusted mortality rate vs. Residential radon levels) NO (MND agemortality rate adjusted forother factors (such as: race, well water use) vs. residential radon levels	1.5
Momcilović B. et al. [37]	Arch higradatoksikol	1999	Croatia	Cases and Controls	29 Cases: 21 Controls: 8 (no smokers)	Analyze thedistribution of ^222^Rn decay products (^210^Po and ^210^Bi) in the lipid and protein fractions of the cortical gray matter and subcortical white matter of the frontal and temporal lobes of the brain of died individuals who had suffered from AD, PD, smokers and no smokers	EGε-G ORTEC system	YES The ^222^Rn decay products is mainly found in the protein fraction of the cortical gray matter and subcortical white matter of AD and smoker patients, compared to PD patients, where these products would accumulate in the lipid fraction of subcortical white matter.	1.5
Lehrer S [51]	Journal of Alzheimer’s Disease	2017	United States	Ecologic	>100	Investigate the possible association between AD mortality rate in USA with radiation exposure (both total and the radon respective one)	“*Assessment of Variations in Radiation Exposure in the United States”* [52], and from article nº 160 - Ionizing Radiation Exposure of the Population of the United States [53]	YES (AD mortality rate vs. radon background radiation and total background radiation in USA)	1.5
Santos, N.V.d. et al. [54]	Scientific Report	2020	Brazil	Descriptive	30	Analyze the levels of emitted α particles by ^210^Po, a ^222^Rn decay product, in the olfactory epithelium, olfactory bulb, frontal lobe, and lung tissues in cadavers from the city of Sao Paulo, Brazil.	The ^210^Po determination in human samples was performed by α particle spectrometry	YES The ^210^Po accumulation in the cadavers´ tissues, suggest that these radionuclides could influence the development of certain diseases such as neurodegenerative ones	1.2

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
