# Peer review of "Radon Exposure and Neurodegenerative Disease"

_ijerph, 2020, doi:10.3390/ijerph17207439_

Round 1

Reviewer 1 Report

There have been many studies conducted to better understand various neurodegenerative diseases. This review paper tried to look into association between radon and neurodegenerative diseases. 

Design, methods and assessments seem reasonable. As correctly assessed, no statistical significance can be made. However, some observations might tempt researchers to dig further. 

major

Overall, this is an interesting read. However, major consideration of syntax and organization of sentences and paragraphs, for

1:Abstract

2:Introduction 

3: Materials and Methods, should be made. 

minor

Also, there are few minor concerns that should be addressed:

1. Line 24: Explain EMN and EA. 

2. Line 25: Explain Rn222, Po210, Bi210.

3. Line 156: Equation should be QI not IQ. 

Author Response

REVIEWER 1:

1.- English has been revised as requested, throughout the text of the article, improving the syntax and understanding of several confusing sentences, and not only in summary, Introduction and Material and Methods.

2.- The definitions of chemical elements and their corresponding symbols have been introduced, and the abbreviations that had remained in Spanish have been corrected (MND, AD or QI, for example)

 The names of the chemical elements have been entered and then their symbol in parentheses, Ex: Uranium-238 (238U) They are highlighted in red.

Lines 27-28, 85-88, 92, 98.

The abbreviations that remained in Spanish have been defined and changed to their corresponding in English: Ex: MND and AD:

Lines 26,48,51,54,59,60,63,67,72,73,106,195,201,278,282,285,294,302,389.

Thank you very much for your work as a reviewer. We greatly appreciate it.

Reviewer 2 Report

Review:" Radon exposure and neurodegenerative disease"

This appears to be a thorough review of the publications dealing with Radon and the relationship with neurological diseases. The approach uses the key data banks for publications related to this topic.

This is an interesting review and study of a literature search related to Radon and disease to see if there is a correlation to effects of radon. The study is well described  and is able to be replicated if desired.

The search of the literature hit all the main data bases (PubMed (Medline), Cochrane, BioMedCentral and Web of Science). The flowchart is well presented.

Being cautious with the data presented and interruption with the articles is an asset of the review.  This review will stimulate some more direct studies of the effects of radon with neurological diseases.

Studies in rodents with given exposures of radon might be an area which could be commented on to neurological disorders. Of course, some of the diseases are not as easily related to rodents but key aspects such as altered behaviors and neurological lesions or myelination issues are likely to be observed with large exposures to any radioactive compounds. Of course dosage and time of exposure are key to the effects. So maybe some future directions and comments on an approach to examine more defined studies in animal models would be helpful

The outlined figure in the approach is very helpful and the table of references used is helpful.

Personally, I learned a lot by reading this review.  

It is nice to see the reviewers did not over state the findings and were cautious in interpretations of the correlations.

I did not see any needed changes in the manuscript except a minor change.

Minor:

Abstract- define "EMN and EA mortality"

I think it might be the conversion to PDF but a place where words are not spaced

"Figure 1. Search summaryforalltearms combinations."

Author Response

REFEREE 2:

1.- The suggestion to explore the possible effects of radon in rodents is appreciated, of which we believe there is not much literature. If there is any on other animals, such as cats and dogs but very limited. Although collateral, we will try to find information about it.

2.- EMN must be MND, Motor Neuron Disease and AD is AD, Alzheimer's Disease.

We sincerely appreciate your work as a reviewer. Thank you very much.

Round 2

Reviewer 1 Report

Dear Authors,

Great work!